# Analysis of Tail Dependence between Sovereign Debt Distress and Bank Non-Performing Loans

**Li Liu [1], Yu-Min Liu [1],*, Jong-Min Kim [2], Rui Zhong [1] and Guang-Qian Ren [1]**

[1]  Business School, Zhengzhou University, Zhengzhou 450001, China; smile_liliu@163.com (L.L.);
    rui.zhong@uwa.edu.au (R.Z.); rgq1982@163.com (G.-Q.R.)
[2]  Division of Science and Mathematics, University of Minnesota-Morris, Morris, MN 56267, USA;
    jongmink@morris.umn.edu
*   Correspondence: yuminliu@zzu.edu.cn; Tel.: +86-136-2384-7252

**Abstract:**  We investigate the tail dependence between sovereign debt distress and bank non-performing loans (NPLs) using a large sample of developed and emerging countries in recent decades. Considering the feedback loop of sovereign debt and bank loan distress, we use three copula models to analyze the asymmetry of tail dependence structure between sovereign debt exposure and bank NPLs. We use the Gaussian copula marginal regression to control the concurrent impact of other macroeconomic variables. We provide evidence that sovereign debt indicates an important determinant of NPLs. We also find that there is tail dependence between sovereign debt distress and bank NPLs, whereas the tail dependence coefficients vary across countries. Our findings shed light on the influence of fiscal distress on bank loan distress and provide immediate implications for the design of macro prudential and financial policy.

**Keywords:**  non-performing loans; sovereign debt distress; tail dependence; gaussian copula regression

**JEL Classification:** H63 G23

## 1. Introduction

The sovereign debt crises in recent decades highlight the influence of sovereign debt distress on the fragility of bank loans. For example, the Russian government's suspension of debt payments in 1998 triggered a dramatic increase of bank bank non-performing loans (NPLs) in Russian banks. The downgrading of Greece's sovereign debts in 2010 raised the ratio of bank NPLs in Greek and other related banks that hold a significant amount of Greece's sovereign debts. Outside of Europe, a similar influence of sovereign debt distress on the performance of bank NPLs also occurs in Argentina, Ecuador, Pakistan, and Ukraine [1]. In the opposite direction, the banking crisis are also important predictors of sovereign debt crises [2]. There is an increasing amount of literature to investigate this perverse feedback loop of sovereign debt distress and bank distress [3–5]. There are several channels that sovereign credit could have a significant impact on the banking system. First, the euro area is linked by the Eurosystem's collateral framework, the joint monetary policy transmission mechanism, and the shared defalt risk of Eurozone countries through the European Stability Mechanism (ESM) and the European Financial Stability Fund (EFSF) [4]. Sovereign debt impacts the financial sector by influencing the balance sheet of financial institutions, lowering the rating of domestic banks, reducing the value of collateral and increasing their financing costs [6]. All the related literature focus on examining the impact of sovereign default (or crises) on bank default (or crises). Since the actual sovereign (bank) defaults or crises are extreme events and occur rarely in history, the sample size of empirical studies is

limited. Meanwhile, there are many concurrent events that occur around sovereign defaults or crises. Since the end of 2009, the increase of higher sovereign risk has pushed up the cost and affected the financing structure of some euro area banks. The banks of Greece, Ireland, and Portugal find it difficult to raise large-scale debt and deposits. It is very difficult to isolate the actual impact of sovereign debt distress on bank loan distress from these concurrent events.

In this study, we adopt three copula models to analyze the tail dependence between sovereign debt distress and bank loan performance. Specifically, we use the ratio of government debt to Gross Domestic Product (GDP) to measure sovereign debt exposure and the ratio of non-performing loans to proxy for bank loan performance. The right tail of sovereign debt exposure and bank NPLs reflects the likelihood of sovereign debt and bank loan distress, respectively. In contrast to the related studies [2–5] that use actual sovereign debt defaults (or crises), our measures mitigate the impact of other concurrent events around actual fiscal or financial crises and also significantly increase the sample size in term of time span and countries. In particular, our study covers 25 countries during a period from 2006 to 2017.

First, a Granger causality test was applied to examine the causality between sovereign debt and bank NPLs. We find that most of the significant causality relationship occurs in the first-year lag variables. For instance, the first-year lag of sovereign debt ratio significantly causes bank NPLs in many countries, such as Cyprus, Portugal, Spain, Germany, Slovakia, Denmark, South Africa, etc. Furthermore, we use Kendall's tau as an alternative measure to exhibit the relationship between each pair of countries in our sample. In particular, we divide our countries into two groups: European countries and BRICS countries (Brazil, Russia, India, China and South Africa). We find that the mean of Kendall's tau among the European countries is about 0.4742, while the mean among the BRICS countries is only 0.0141. This evidence suggests that geographical location is one of the factors affecting the spillover effects of sovereign debt on bank NPLs.

To analyze the asymmetry of the tail dependence structure between sovereign debt and bank NPLs, we employ three of the most frequently used copula models in finance [7]: Student's *t*-copula (symmetric association of tail dependence), rotated Clayton copula (upper-tail dependence), and Joe copula (upper-tail dependence). We employ these copulas because they are able to analyze the upper tail dependence. Our empirical results suggest that the tail dependence between sovereign debt and bank NPLs varies across countries. Most tail dependence results are consistent across the three copula approaches with slight variation, especially in the *t* copula. All copula approaches point out that the highest tail dependence between sovereign debt ratio and bank NPLs ratio occurs in Ireland and Croatia. In addition, the tail dependence is high in Denmark, Belgium, Cyprus, Latvia, Austria, and Portugal in contrast to other countries in our sample.

Last, since the bank NPLs ratio is affected by many factors simultaneously, we use the Gaussian copula regression method (GCRM) to isolate the important impact of sovereign debt on bank NPLs from other known determinants. Particularly, we control GDP, inflation rate, government fiscal expenditure and government fiscal revenue. We document significant and positive relation between sovereign debt ratio and bank NPLs ratio in 15 out of 25 countries, which adds more credence to the upper tail dependence between sovereign debt distress and bank non-performing loans in some countries, such as Ireland and Croatia.

This research contributes to the related literature from at least two aspects. First, we use a new method to analyze the upper tail dependence between sovereign debt and bank NPLs. Most closely related literature uses classical regression models to examine the relation between sovereign debt defaults and financial crises [8–11]. In contrast to the classical regression models, copula models have an advantage of analyzing the tail dependence between two distributions. For instance, Reboredo and Ugolini use the CoVaR-copula approach and vine-copula conditional VaR approach to analysis the impact of the European debt crisis on the banking risk in debt markets [12,13]. In this study, we use three alternative copula approaches to explore the tail dependence between sovereign debt ratio and bank NPLs ratio, which deepens the understanding of this relation.

Second, our study enriches the literature on the determinants of bank NPLs by providing new evidence from the aspect of macroeconomic factors. There is a long history and large amount of literature to examine the determinants of bank NPLs. Most of the identified determinants are microeconomic factors, mostly bank characteristics, such as cost efficiency [14]), bank's risk management function [15], government ownership [16], bank credit growth [17], etc. Recently, especially after the sub-prime financial crises, a strand of literature has emerged to examine the impact of macroeconomic and policy-related factors on bank non-performing loans, including monetary policy [18], inflation rate [19,20], GDP [10,21,22], fiscal expenditure and revenue [23–26], unemployment rates and housing price index [27], and policy rates [28,29]. Makri et al. reveal strong correlations between NPLs and public debt of Eurozone's banking systems for the period 2000–2008, and fiscal problems may raise bad loans in this region [11]. Ghosh finds that liquidity risk, greater capitalization, greater cost inefficiency, poor credit quality, and banking industry size significantly increase bank non-performing loans. Similarly, inflation, state unemployment rates, and US public debt significantly increase bank non-performing loans [27]. In contrast to previous studies, this paper identifies a new macroeconomic factor, sovereign debt to GDP ratio, which affects bank NPLs ratios. Empirical evidence shows that the impact of sovereign debt ratio on bank non-performing loans varies across countries. Liu et al. find that the government debt and NPLs of EU and BRICS countries increased drastically after the crisis, and crisis countries are contagious [30].

The organization of this paper is as follows: Section 2 describes the econometric methodology in the analysis. Section 3 defines the variables and data descriptions. Section 4 discusses the empirical results. Finally, Section 5 concludes.

## 2. Econometric Methodology

### 2.1. Granger Causality Tests

Granger causality shows that the lagged value of one variable is conducive to predicting another variable [31]. To study the causality relationship between sovereign debt and bank NPLs, we follow the Granger causality test to analyze the existence of a causality relationship. More precisely, we plan to test whether the change of sovereign debt precedes NPLs or, on the contrary, whether NPLs precede the sovereign debt (and even whether these relationships are bidirectional).

Consider two variables, $X_t$ and $Y_t$, $X_t$ Granger-causes $Y_t$ suggested that lags of $X_t$ provide useful information to explain present values of $Y_t$ and vice versa. Under these conditions, the Granger causality model is as follows:

$$X_t = \varphi_1 + \sum_{i=1}^{k} a_{1i} X_{t-i} + \sum_{j=1}^{k} b_{1j} Y_{t-j} + \varepsilon_{1t} \tag{1}$$

$$Y_t = \varphi_2 + \sum_{i=1}^{k} a_{2i} X_{t-i} + \sum_{j=1}^{k} b_{2j} Y_{t-j} + \varepsilon_{2t} \tag{2}$$

where $X$ and $Y$ denote NPLs and sovereign debt, interchangeably; $\varphi_1$ and $\varphi_2$ are the intercepts of the equation; and $k$ is the lag length. $a$ and $b$ represent estimation coefficients, and $\varepsilon$ denotes error terms. This model is suitable for testing the causality relationships between NPLs and sovereign debt during 2006–2017. For each sample country, alternative causal relationship can be found [32]. There is a one-way Granger-causality from $X$ to $Y$ if not all $b_{2j}$ are zero, but all $a_{1i}$ are zero. Similarly, there is a one-way Granger-causality from $Y$ to $X$ if not all $a_{2i}$ are zero, but all $b_{1j}$ are zero. Additionally, there is a two-way Granger-causality between $X$ and $Y$ if all $b_{2j}$ and $a_{2i}$ are not zero. There is no Granger-causality between $X$ and $Y$ if $b_{2j}$ and $a_{2i}$ are zero [33].

## 2.2. Kendall's Tau Coefficient

In statistics, Kendall's tau coefficient is a measure of nonlinear dependence between two random variables $X$ and $Y$, which was put forward by Maurice Kendall in 1938. Kendall's tau shows that there is a monotonic (but not necessarily linear) relationship between the two variables. A Kendall's tau test is a nonparametric hypothesis test based on the tau correlation coefficient [34].

According to Kendall, suppose a pair of points $(X_i, Y_j)$ and $(X_j, Y_j)$ are considered to be concordant if $X_i < X_j$ and $Y_i < Y_j$ or if $X_i > X_j$ and $Y_i > Y_j$, meaning that the pairs are in the same order to each variable. Namely, a higher value of $X$ corresponds to a higher value of $Y$; on the contrary, a lower value of $X$ corresponds to a lower value of $Y$. The pairs are considered to be discordant if $X_i < X_j$ and $Y_i > Y_j$ or if $X_i > X_j$ and $Y_i < Y_j$, meaning that the values of variables are arranged in opposite directions; in other words, a lower value of $X$ corresponds to a higher value of $Y$, and a higher value of $X$ corresponds to a lower value of $Y$ [35,36]. The pairs are tied if $X_i = X_j$ and/or $Y_i = Y_j$.

If two random variables $X$ and $Y$ obey joint distribution $H(x, y)$ and two vectors of $(X_i, Y_i)$ and $(X_j, Y_j)$ are independent, then Kendall's tau is defined by

$$\tau = P[(X_i - X_j)(Y_i - Y_j) > 0] - P[(X_i - Y_j)(X_i - X_j) < 0] \tag{3}$$

Equation (3) will calculate a value in $[-1,1]$, which is the same as Pearson's correlation. The higher the absolute value of $\tau$, the stronger the correlation between the two variables. Specifically, a positive value indicates that the higher value of one variable is associated with the higher value of another variable, and a negative value is the opposite [37].

## 2.3. Copula Function and Tail Dependence

Copulas were first introduced in the Sklar theorem and further developed by Joe (1997). A copula is a function that relates univariate distribution to one-dimensional marginal multivariate distribution of related variables

$$H(x, y) = C(F_X(x), F_Y(y)) \tag{4}$$

where $H(.)$ is a bivariate function, and $F_X(.)$ and $F_Y(.)$ are cumulative distribution functions of $X$ and $Y$. Sklar's theorem says that univariate margins can be separated from the dependence structure, and the dependency structure can be represented by a copula, which is $C$ [38]. A bivariate copula is a function $C: [0,1]^2 \rightarrow [0,1]$, which has the following properties, that is, $u = F_X(x)$ and $v = F_Y(y)$:

(1)  The domain of $C(u, v)$ is $[0,1] \times [0,1]$
(2)  $C(u, 1) = C(1, u) = u$, $C(v, 1) = C(1, v) = v$, $\forall u, v \in [0,1]$
(3)  $C(u, v)$ has zero fundamentals and is incremented in two dimensions

As can be seen from the above definition, copulas are a useful way to model dependent random variables. Copulas have some certain properties that are very useful in dependency studies. Firstly, copulas are always constant for strictly increasing the transformation of random variables. Secondly, consistency measurements between widely used random variables, such as Kendall's tau and Spearman's rho, are attributes of copulas. Thirdly, tail dependence is also an attribute of copulas.

Nelsen defines the upper and lower tail dependence coefficients for $\tau^U \in [0, 1]$ and $\tau^L \in [0, 1]$ of $(X, Y)$ [39]

$$\begin{aligned} \tau^U &= \lim_{\varepsilon \to 1} P[U > \varepsilon | V > \varepsilon] = \lim_{\varepsilon \to 1} P[V > \varepsilon | U > \varepsilon] \\ \tau^L &= \lim_{\varepsilon \to 0} P[U \leq \varepsilon | V \leq \varepsilon] = \lim_{\varepsilon \to 1} P[V \leq \varepsilon | U \leq \varepsilon] \end{aligned} \tag{5}$$

The upper (right) and lower (left) tail dependence of a normal copula are equal, which is $\tau^L = \tau^U = 0$, and this means that variables are independent in the extreme of distribution. The most common assumption is the normal copula in finance although it does not have tail dependence. There are many types of copulas that have left or right tail dependence.

### 2.3.1. Student's *t* Copula

Similar to the normal distribution, the *t* distribution is a symmetrical bell-shaped distribution, but its tail is heavier, which means it is more likely to produce values far below its mean [40].

$$C^t(u,v|\rho,\lambda) = \int_{-\infty}^{T_\lambda^{-1}(u)} \int_{-\infty}^{T_\lambda^{-1}(v)} \frac{1}{2\pi\sqrt{1-\rho^2}} \left[1 + \frac{s^2 + r^2 - 2\rho rs}{\lambda(1-\rho)^2}\right]^{-\frac{\lambda+2}{2}} dsdr \tag{6}$$

where $\lambda$ denotes the degrees of freedom of the t copula, $\rho$ is the coefficient of linear correlation, and $T_\lambda^{-1}$ is the inverse of the standard univariate *t*-distribution function with $\lambda$ degrees of freedom. The Kendall's tau of the *t* copula is $\frac{2\arcsin\rho}{\pi}$, and

$$\begin{aligned} \tau^L &= 2 - 2t_{k+1}\left(\frac{\sqrt{k+1}\sqrt{1-\rho}}{\sqrt{1+\rho}}\right) \\ \tau^U &= 2 - 2t_{k+1}\left(\frac{\sqrt{k+1}\sqrt{1-\rho}}{\sqrt{1+\rho}}\right) \end{aligned} \tag{7}$$

where $t_{k+1}\left(\frac{-\sqrt{k+1}\sqrt{1-\rho}}{\sqrt{1-\rho}}\right)$ represents the value of the standard t-distribution function with the degree of freedom $k+1$ at $\left(\frac{-\sqrt{k+1}\sqrt{1-\rho}}{\sqrt{1-\rho}}\right)$.

### 2.3.2. Clayton Copula

Clayton proposed one of the first bivariate association models for survival analysis [41]. Especially, the Clayton copula is a popular method for time dependent association data, and it has a lower tail dependence between variables and can fit structural relationships with lower tail dependence well; it is higher for negative events than joint positive events.

$$C(u,v|\theta) = (u^{-\theta} + v^{-\theta} - 1)^{-1/\theta} \tag{8}$$

where $\forall \theta \in (0,\infty)$, the Kendall's tau of the Clayton copula is $\frac{\alpha}{2+\alpha}$, and $\tau^L = 2^{-1/\alpha}$.

### 2.3.3. Joe Copula

In order to estimate the copula from a bivariate observational data set, Joe proposed the tail dependence concept with copulas [42]. It relates the amount of dependence of the upper-right quadrant tail or the lower-left quadrant tail of a bivariate distribution. The upper tail dependence that can be found by the Joe copula is defined by

$$C(u,v|\theta) = 1 - \left[(1-u)^\theta + (1-v)^\theta - (1-u)^\theta(1-v)^\theta\right]^{1/\theta} \tag{9}$$

where $\theta \in [1,\infty)$.

### 2.4. Gaussian Copula Regression Method

Linear regression has been extensively used by statisticians to find the relationship between dependent variables [43]. However, the assumption of linear relationship is too unrealistic and restrictive in real application. Copula regression is more suitable than a linear model when the dependent variable does not follow a normal distribution. Gaussian copula regression has been increasingly popular in longitudinal data analysis, time series, mixed data, and spatial statistics. Copula regression uses various copula models to represent the joint distribution of a pair of continuous and discrete random variables. The marginal can be defined by various parametric distributions. The marginal distributions are fitted to the bivariate joint distribution by maximum likelihood [44,45].

To avoid the multicollinearity in regression analysis, we employ the Gaussian copula marginal regression (GCMR) for each country to measure the macroeconomic variables to assess the impact on NPLs.

Consider a vector of $n$ dependent variables $Y_1, \ldots, Y_n$. Let $F(\cdot|x_i)$ be the marginal cumulative distribution of $Y_i$ for covariates $x_i$. The joint data cumulative distribution function in the Gaussian copula regression is expressed as

$$\Pr(Y_1 \leq y_1, \ldots, Y_n \leq y_n) = \Phi(\tau_1, \ldots, \tau_n; P) \tag{10}$$

where $\tau_i = \Phi^{-1}\{F(y_i|x_i)\}$. For the correlation matrix of P, $\Phi(\cdot)$ represents the univariate standard normal cumulative distribution function and $\Phi(\cdot; P)$ denotes the multivariate standard normal cumulative distribution function. More details of the Gaussian copula model are described in Song [46] and Masarotto [47]. In particular, an equivalent formulation of the Gaussian copula is given by

$$Y_i = h(x_i, \tau_i) \tag{11}$$

where $\tau_i$ represents a stochastic error. The Gaussian copula regression model assumes that

$$h(x_i, \tau_i) = F^{-1}\{\Phi(\tau_i)|x_i\} \tag{12}$$

and the vector of errors $\tau = (\tau_1, \ldots, \tau_n)^T$ follows multivariate normal distribution.

To avoid linearity, normal and independent assumptions, Gaussian marginal distributions for individual predictor variables are used with Gaussian copula functions in this paper.

$$\begin{aligned} \text{NPLs ratio}_t = \quad & \beta_0 + \beta_1 \times \text{government} - \text{to} - \text{GDP ratio}_t + \beta_2 \times \text{Expenditure}_t + \\ & \beta_3 \times \text{GDP}_t + \beta_4 \times \text{Revenue}_t + \mu_t \end{aligned} \tag{13}$$

where $\mu_t$ is the error term, $\mu_t = \varphi \mu_{t-1} + \omega_t$, and $\omega_t \sim iidN(0, \sigma_\omega^2)$ for the GCMR.

## 3. Definition of Variables and Data Description

### 3.1. Variables and Data Description

In this paper, we investigate tail dependence between sovereign debt distress and bank NPLs. The data are comprised of general government gross debt to GDP ratio, bank NPLs, GDP, government fiscal expenditure, government fiscal revenue, and inflation rate which are retrieved from Eurostat and the International Monetary Fund (IMF). Table 1 presents macroeconomic variables in this study and their corresponding sources. We exclude these countries in the sample because data availability problems in some countries, such as Luxembourg, the Netherlands, and Finland. The final sample includes 30 countries, including 25 EU countries and 5 BRICs countries. This paper covers the period of 11 years (2006–2017), which includes both the global financial crisis and European sovereign debt crisis period.

As can be seen from Figures 1 and 2, there are similar fluctuations of two key variables between most sample countries. Meanwhile, the rise of the bank non-performing loan ratio is often accompanied by the expansion of general government gross debt. This feature provides evidence that a dependence of two key variables may exist.

**Table 1.** Definition of variables used in this paper.

| Variable | Definition | Source |
|---|---|---|
| Non-performing loans (NPLs) | The ratio of non-performing loans to total loans. | Eurostat, International Monetary Fund (IMF) |
| Sovereign debt | The general government gross debt to GDP ratio. | Eurostat, IMF |
| Gross domestic product (GDP) | GDP has a significant negative impact on NPLs, because the growth of GDP creates more jobs, which increases income of borrowers and reduces NPLs. Therefore, the level of NPLs will rise when the economy slows down. | IMF |
| Inflation rate | It is represented by a percentage change in regional consumer price index (CPI) and low liquidity level is conducive to economic growth, while high liquidity rate weakens borrowers' solvency by reducing their real income, thus increasing NPLs. | IMF |
| Government fiscal expenditure | Government fiscal expenditure refers to the funds expended by the government. Economic growth through fiscal expenditure and government investment, as well as more central government deficits and money supply has been greatly limited after the economic crisis. Meanwhile, insufficient fiscal revenue will offset the growth of tax revenue. When the government faces a budget deficit, it will generate public debt. | IMF |
| Government fiscal revenue | Government revenue is the income available to fund the activities of a government. High fiscal revenue usually means the government controls a large share of financial resources has the ability to repay bank loans, while fiscal distress implies that fiscal revenue cannot satisfy government's expenditures. | IMF |

From Figure 1, it is obvious that the NPLs ratio of the sample countries experienced a huge fluctuation during 2007–2017. After the global financial crisis of 2008, the bank non-performing loan ratio changed significantly for many EU countries such as Greece, Cyprus, Lithuania, Ireland, Latvia, Italy, Croatia, Bulgaria, Hungary, Portugal, Malta, Romania, et al. Meanwhile, the scale of the NPLs ratio expanded significantly when the European sovereign debt crisis emerged. After the debt crisis' peak in 2013, the NPLs ratio started to decrease in most sample countries. One of the important events of the Greek crisis occurred on 18 October 2009 when the Greek government announced that the budget deficit had increased to at least 12% of the GDP, double the government's estimate. The NPLs ratio of Cyprus and other EU countries rose sharply when the Greek government bonds defaulted, as those countries' banks invested heavily in Greek sovereign debt. Therefore, the economic crisis did spread to countries such as Portugal, Italy, Ireland, Spain, France, and other countries with strong economic strength in the Eurozone. However, the NPLs ratio for the non-crisis countries (Estonia, Sweden, Germany, China, Brazil, South Africa, et al.) remained relatively stable after the outbreak of the crisis. It cannot be ignored that the non-performing loan ratio of the banks in two BRICS countries (India and Russia) has been continuously increasing.

Figure 2 shows the government-to-GDP ratio for each of the sample countries in the sample. The evolutions of these ratios are very similar; thus, we can distinguish some different periods marked by the global financial risk of 2008 and the European sovereign debt crisis of 2009. It can be seen that the government-to-GDP ratio of most sample countries is lower than the 90% debt cliff before 2007. As the global financial crisis deepened, the government-to-GDP ratio began to largely increase in all of the sample countries, especially in Greece, Italy, Portugal, Ireland, and Belgium. By the end of 2017, the government-to-GDP ratio of Greece reached 180.8%, three times the Eurozone government-to-GDP ratio limit set at 60%. Lately, Belgium, Spain, Cyprus, Slovenia, and the United Kingdom also show large increases. Different from the bank non-performing loan ratio, the government-to-GDP ratio of the BRICS countries such as China, Brazil, and South Africa has grown slowly since 2008. However, the government-to-GDP ratio of Brazil and India has reached above 69%, higher than some EU countries (Sweden, Slovakia, Lithuania, et al.). Large government debt in many sample countries (developed economics and underdeveloped economics) has become a serious problem.

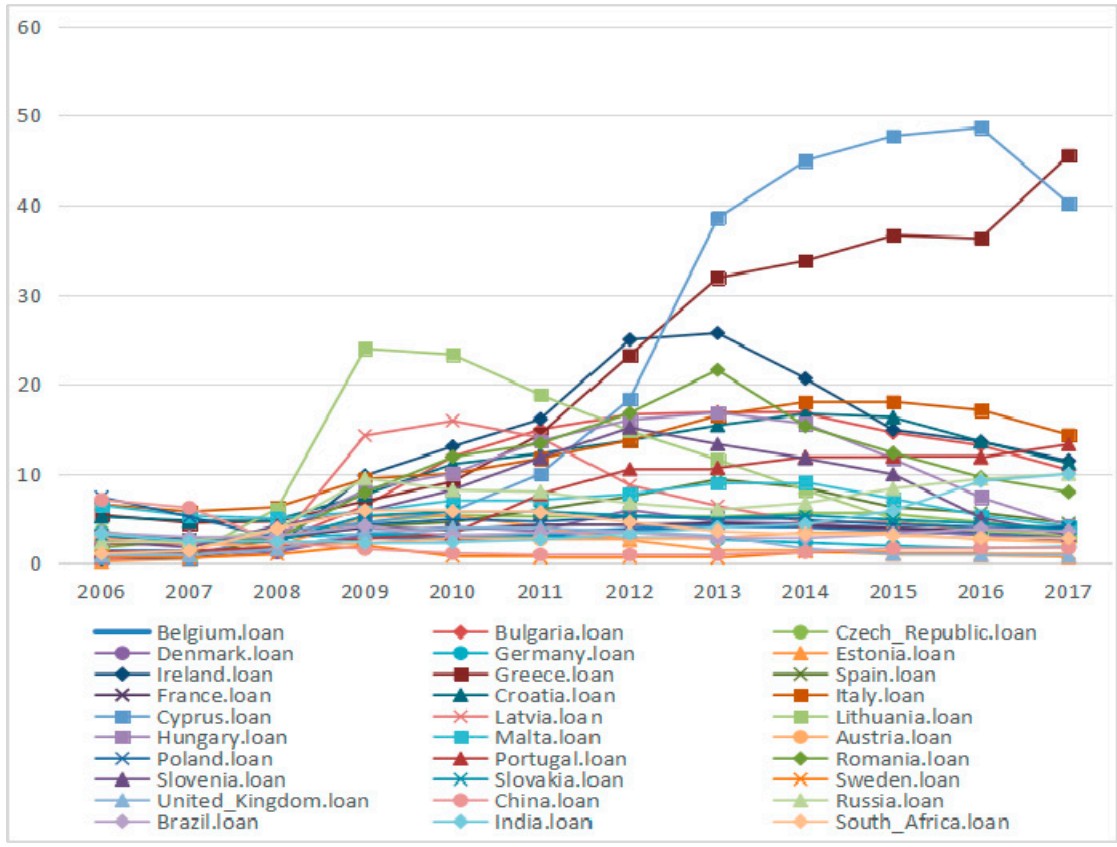

**Figure 1.** Bank non-performing loans (NPLs) ratio of the sample countries since during 2007–2017. Source: Eurostat; International Monetary Fund.

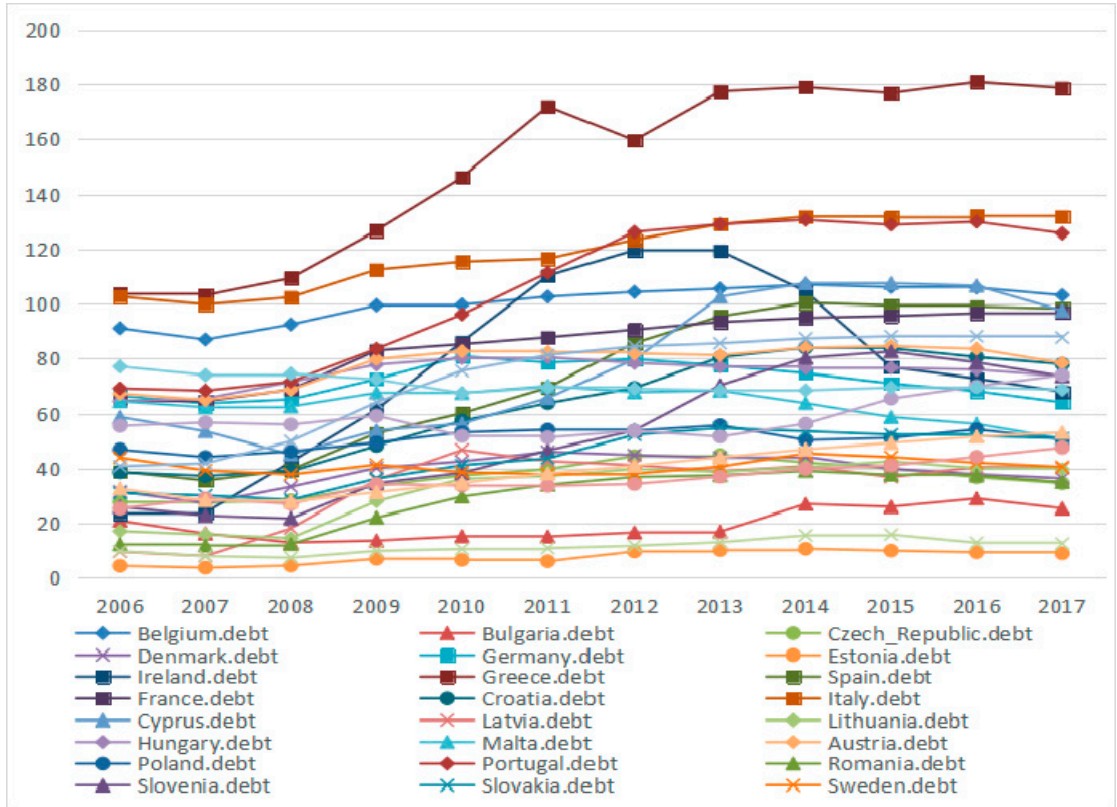

**Figure 2.** Government-to-gross domestic product (GDP) ratio of the sample countries since during 2007–2017. Source: Eurostat; International Monetary Fund.

### 3.2. Descriptive Statistics

Basic summary descriptive statistics of key variables used in this paper are presented in Tables 2 and 3. The mean is significantly different from zero for the NPLs ratio and government-to-GDP ratio in the sample countries. As shown, the average of some sample countries is higher than other sample countries. The reason is simply that as the European sovereign debt crisis unfolded, some countries with weak competitiveness and loose financial supervision—such as Greece, Cyprus, Ireland, etc.—widened the means in Tables 2 and 3 more speedily than did BRICS countries. Both the minimum/maximum and the standard deviations indicate that there is a notable time series variation in the key variables. For example, the bank non-performing loan ratio for Greece reached a maximum of 45.57 basis points (see Table 2), and the government-to-GDP ratio for Portugal reached a maximum of 130.6 basis points (see Table 3).

In the same time period, the NPLs ratio for Cyprus and the government-to-GDP ratio for Greece reached maximum values of 48.68 basis points and 180.8 basis points, respectively. Meanwhile, the mean values of the key variables are typically very close to the average values. As shown in Table 2, negative values for skewness are more pronounced for the Czech Republic than for the other sample countries, which suggests a bigger probability of large decreases, suggesting that those distributions have long left tails. Meanwhile, there is evidence of positive skewness for Sweden, China, India, Latvia, and Lithuania and therefore distributions with long right tails. Considering that the kurtosis of a normal distribution is generally 3, Table 2 shows that most the distribution of kurtosis of the NPLs ratio is lower than 3, suggesting that it does not have a heavy-tailed distribution.

**Table 2.** Descriptive statistics for the bank non-performing loan ratio.

| | Mean | Median | Minimum | Maximum | St.D | Skewness | Kurtosis |
|---|---|---|---|---|---|---|---|
| Belgium | 2.9642 | 3.1900 | 1.1600 | 4.2400 | 1.0694 | −0.6117 | 2.0734 |
| Bulgaria | 10.7067 | 12.545 | 2.1000 | 16.8800 | 5.9127 | −0.4996 | 1.6586 |
| Czech Republic | 4.4858 | 4.8950 | 2.3700 | 5.6100 | 1.1010 | −0.7795 | 2.2255 |
| Denmark | 3.1392 | 3.4800 | 0.4000 | 5.9500 | 1.6898 | −0.3115 | 2.2255 |
| Germany | 2.6633 | 2.7800 | 1.6900 | 3.4000 | 0.5744 | −0.4537 | 1.9127 |
| Estonia | 2.1083 | 1.4300 | 0.2000 | 5.3800 | 1.8157 | 0.8590 | 2.2676 |
| Ireland | 12.7833 | 13.3300 | 0.5300 | 25.7100 | 8.6392 | −0.0637 | 2.0023 |
| Greece | 21.0450 | 18.8500 | 4.5000 | 45.5700 | 15.1776 | 0.2151 | 1.4648 |
| Spain | 5.0642 | 5.1200 | 0.7000 | 9.3800 | 2.7234 | −0.1510 | 2.1743 |
| France | 3.6867 | 3.8700 | 2.7000 | 4.5000 | 0.6312 | −0.3824 | 1.6509 |
| Croatia | 11.0725 | 11.7350 | 4.7500 | 16.7100 | 4.4540 | −0.3116 | 1.6736 |
| Italy | 12.3108 | 12.7450 | 5.7800 | 18.0600 | 4.6595 | −0.1378 | 1.5490 |
| Cyprus | 21.9842 | 14.1800 | 0.6000 | 48.6800 | 20.1756 | 0.2544 | 1.2686 |
| Latvia | 6.6000 | 4.6200 | 0.5000 | 15.9300 | 5.4205 | 0.6585 | 1.9852 |
| Lithuania | 10.0508 | 7.1350 | 1.0000 | 23.9900 | 8.3663 | 0.5788 | 1.8944 |
| Hungary | 9.3150 | 9.1400 | 2.3000 | 16.8300 | 5.4712 | 0.0334 | 1.5220 |
| Malta | 6.5892 | 6.7450 | 4.1000 | 9.05000 | 1.5497 | 0.1544 | 2.0655 |
| Austria | 2.6900 | 2.7250 | 1.9000 | 3.4700 | 0.4563 | 0.1258 | 2.4942 |
| Poland | 4.7175 | 4.7400 | 2.8200 | 7.4000 | 1.0769 | 0.8988 | 4.8279 |
| Portugal | 7.3750 | 9.1250 | 1.3000 | 13.3000 | 4.8475 | 0.1852 | 1.2605 |
| Romania | 10.3367 | 10.7600 | 1.8000 | 21.6000 | 6.1126 | 0.1340 | 2.1850 |
| Slovenia | 7.7317 | 7.0000 | 1.8000 | 15.1800 | 4.5795 | 0.2129 | 1.6299 |
| Slovakia | 4.4883 | 4.9850 | 2.5000 | 5.8000 | 1.2140 | −0.6285 | 1.9133 |
| Sweden | 0.9758 | 0.9000 | 0.6000 | 2.0000 | 0.3941 | 1.4469 | 4.7866 |
| United Kingdom | 2.1617 | 1.6250 | 0.8100 | 3.9600 | 1.3342 | 0.3073 | 1.3017 |
| China | 2.3125 | 1.6350 | 0.9500 | 7.1000 | 2.0789 | 1.6631 | 4.0742 |
| Russia | 6.8017 | 7.3650 | 2.4000 | 10.0000 | 2.6596 | −0.5731 | 2.0161 |
| Brazil | 3.3600 | 3.3800 | 2.8500 | 4.2100 | 0.4157 | 0.6093 | 2.5971 |
| India | 4.3675 | 3.3350 | 2.3000 | 9.9800 | 2.6547 | 1.2948 | 3.2111 |
| South Africa | 3.6742 | 3.4350 | 1.1000 | 5.9000 | 1.5988 | −0.0098 | 2.0376 |

Table 3 reports summary statistics for the general government gross debt of the sample countries. Average means close to zero, which is like the bank non-performing loan ratio. Negative values for skewness are more pronounced for Latvia, Hungary, and Belgium, suggesting that those distributions have long left tails. Meanwhile, there is evidence of positive skewness for Sweden, Brazil, China, and India and therefore of distributions with long right tails. EU countries (except Sweden) have negative values for skewness, whereas the five BRICS countries have opposite results. The distribution of the kurtosis of the government-to-GDP ratio does not comply with the normal distribution generated from Table 3.

**Table 3.** Descriptive statistics for the government-to-GDP ratio.

|  | Mean | Median | Minimum | Maximum | St.D | Skewness | Kurtosis |
|---|---|---|---|---|---|---|---|
| Belgium | 100.3583 | 102.8500 | 87.0000 | 107.0000 | 6.6701 | −0.8902 | 2.4051 |
| Bulgaria | 19.6333 | 16.8500 | 13.0000 | 29.0000 | 5.7362 | 0.4649 | 1.6080 |
| Czech Republic | 36.4417 | 37.1000 | 27.5000 | 44.9000 | 6.2432 | −0.2141 | 1.7903 |
| Denmark | 39.0333 | 40.0500 | 27.3000 | 46.1000 | 5.9322 | −0.6277 | 2.2842 |
| Germany | 71.9000 | 71.8000 | 63.7000 | 80.9000 | 6.3577 | 0.0684 | 1.5050 |
| Estonia | 7.6083 | 8.0000 | 3.7000 | 10.7000 | 2.5300 | −0.2885 | 1.5525 |
| Ireland | 75.7500 | 74.8500 | 23.6000 | 119.6000 | 34.0440 | −0.2120 | 1.8718 |
| Greece | 151.1000 | 165.8500 | 103.1000 | 180.8000 | 31.9020 | −0.5597 | 1.6123 |
| Spain | 72.8917 | 77.6000 | 35.6000 | 100.4000 | 26.4605 | −0.2630 | 1.4036 |
| France | 85.1750 | 89.2000 | 64.5000 | 97.0000 | 12.4362 | −0.7907 | 2.0593 |
| Croatia | 63.3833 | 66.6000 | 37.3000 | 84.0000 | 18.6716 | −0.2974 | 1.4865 |
| Italy | 119.0583 | 119.9500 | 99.8000 | 132.0000 | 12.6018 | −0.3452 | 1.5921 |
| Cyprus | 77.8750 | 72.7000 | 45.1000 | 107.5000 | 24.8807 | 0.0977 | 1.2836 |
| Latvia | 33.3000 | 39.5500 | 8.0000 | 46.8000 | 13.3813 | −1.1015 | 2.5721 |
| Lithuania | 32.5500 | 38.0000 | 14.6000 | 42.6000 | 10.6798 | −0.8592 | 1.9943 |
| Hungary | 74.8750 | 76.6500 | 64.5000 | 80.5000 | 5.2343 | −1.0305 | 2.8415 |
| Malta | 63.3583 | 64.1500 | 50.8000 | 70.1000 | 5.7175 | −0.9038 | 2.9411 |
| Austria | 78.3167 | 81.6000 | 65.0000 | 84.6000 | 7.0776 | −1.0123 | 2.3322 |
| Poland | 50.8000 | 50.8500 | 44.2000 | 55.7000 | 3.5868 | −0.4349 | 2.0792 |
| Portugal | 105.8917 | 118.5500 | 68.4000 | 130.6000 | 26.2724 | −0.4422 | 1.4574 |
| Romania | 28.8333 | 34.5000 | 11.9000 | 39.1000 | 11.0311 | −0.7188 | 1.7858 |
| Slovenia | 52.4583 | 50.2000 | 21.8000 | 82.6000 | 23.7567 | −0.0003 | 1.3967 |
| Slovakia | 43.8500 | 47.3000 | 28.5000 | 54.7000 | 10.0848 | −0.4233 | 1.5416 |
| Sweden | 40.8500 | 40.6500 | 37.8000 | 45.5000 | 2.6586 | 0.4120 | 1.8766 |
| United Kingdom | 72.9333 | 82.9000 | 40.8000 | 88.2000 | 18.7544 | −0.8410 | 2.0280 |
| China | 35.6000 | 34.3000 | 25.4000 | 47.6000 | 6.7618 | 0.1938 | 2.1800 |
| Russia | 11.5417 | 11.3500 | 7.4000 | 15.9000 | 2.6569 | 0.1562 | 2.2124 |
| Brazil | 58.4508 | 56.1300 | 51.2700 | 74.0400 | 7.4771 | 1.0211 | 2.7483 |
| India | 70.7583 | 69.5500 | 67.5000 | 77.1000 | 3.0125 | 0.9404 | 2.5640 |
| South Africa | 39.9167 | 39.6000 | 27.8000 | 53.1000 | 9.0799 | 0.0783 | 1.5803 |

## 4. Empirical Results and Discussion

### 4.1. Granger Results

Table 4 reports the *p*-values for Granger causality between NPLs and sovereign debt at various levels of lags in the 30 countries. We observe a bi-directional causality relationship in Cyprus, Italy, Portugal, Romania, Spain, Denmark, Sweden, and South Africa. It cannot be ignored that the bi-directional causal link is mainly found in northern and southern European countries. In those countries, the evidence for bi-directional causality is consistent with the evolution of sovereign debt and NPLs. A substantial increase in the size of sovereign debt occurred in the outbreak of the financial crisis, suggesting a strong interaction between sovereign debt and NPLs.

It can also be observed that unidirectional causality occurs? between NPLs and sovereign debt for Bulgaria, Greece, Malta, Slovenia, Ireland, United Kingdom, Austria, Czech Republic, Germany, Lithuania, Brazil, and India. Our results show that there are several countries for which the null hypothesis of causality between sovereign debt and NPLs cannot be rejected, including Greece, Malta, Slovenia, Ireland, Austria, Czech Republic, Germany, Lithuania, Brazil, and India. The debt ratio rises significantly in the European countries after the recent European debt crisis. Considering the obviously low return on debt accumulation, it seems to have exacerbated the scale of NPLs. Note that Brazil and India are emerging countries with a strong willingness to lend. Nevertheless, we find that bank NPLs

cannot significantly cause the change of sovereign debt in Bulgaria and the United Kingdom at the conventional level.

    A causal relationship exists between sovereign debt and NPLs since the non-causality null hypothesis is rejected in Croatia, Belgium, France, Hungary, Poland, Estonia, Latvia, China, and Russia. Statistical evidence exists against the null hypothesis of an absence of causal relationship between sovereign debts and debt in sample countries, especially in Western Europe, central Europe, and Eastern European countries. It is important to recall that their economic problems have been merged after the economic crisis and sovereign debt crisis.

**Table 4.** Granger causality test for the lag $k_i$ ($i$ = 1,2,3).

| Country | Debt → NPLs | | | NPLs → Debt | | |
|---|---|---|---|---|---|---|
| | $k = 1$ | $k = 2$ | $k = 3$ | $k = 1$ | $k = 2$ | $k = 3$ |
| Belgium | 0.0690 | 0.0928 | 0.1287 | 0.0621 | 0.06772 | 0.2773 |
| Bulgaria | 0.1033 | 0.4841 | 0.9006 | 0.0085 ** | 0.0767 | 0.3815 |
| Czech Republic | 0.0398 * | 0.0362 * | 0.1691 | 0.1746 | 0.2878 | 0.2525 |
| Denmark | 0.0056 ** | 0.598 | 0.9068 | 0.0001 *** | 0.0133 * | 0.0907 |
| Germany | 0.0011 ** | 0.1085 | 0.0445 * | 0.0928 | 0.0587 | 0.2920 |
| Estonia | 0.4735 | 0.2668 | 0.1499 | 0.1239 | 0.3308 | 0.1546 |
| Ireland | 0.7105 | 0.0138 * | 0.2595 | 0.2626 | 0.6566 | 0.7956 |
| Greece | 0.0102 * | 0.0347 * | 0.1988 | 0.9357 | 0.5671 | 0.2092 |
| Spain | 0.0028 ** | 0.5248 | 0.1932 | 0.0024 ** | 0.0358 * | 0.1492 |
| France | 0.4955 | 0.8559 | 0.6277 | 0.1203 | 0.2864 | 0.7300 |
| Croatia | 0.1952 | 0.1818 | 0.1523 | 0.9109 | 0.8177 | 0.7420 |
| Italy | 0.3479 | 0.0128 * | 0.1527 | 0.1608 | 0.0003 *** | 0.2963 * |
| Cyprus | 0.0003 *** | 0.5192 | 0.6840 | 0.0042 ** | 0.0109 * | 0.8223 |
| Latvia | 0.9098 | 0.3147 | - | 0.8794 | 0.3847 | 0.0400 * |
| Lithuania | 0.0089 ** | 0.0042 ** | 0.0638 | 0.1923 | 0.3646 | 0.6983 |
| Hungary | 0.6667 | 0.9697 | 0.6400 | 0.4286 | 0.2949 | 0.3772 |
| Malta | 0.0177 * | 0.0599 | 0.3195 | 0.02565 | 0.6050 | 0.2901 |
| Austria | 0.01399 * | 0.0352 * | 0.031 * | 0.1153 | 0.3596 | 0.8432 |
| Poland | 0.4803 | 0.6403 | 0.6308 | 0.6490 | 0.2583 | 0.4747 |
| Portugal | 0.0014 ** | 0.2132 | 0.4841 | 0.0279 * | 0.4442 | 0.8346 |
| Romania | 0.0245 * | 0.1286 | 0.5888 | 0.036 * | 0.0515 | 0.3409 |
| Slovenia | 0.4660 | 0.2373 | 0.0443 * | 0.8733 | 0.8929 | 0.8374 |
| Slovakia | 0.0089 ** | 0.0171 * | 0.1097 | 0.2338 | 0.7354 | 0.7900 |
| Sweden | 0.0044 ** | 0.3647 | 0.8736 | 0.0052 ** | 0.1596 | 0.0232 * |
| United Kingdom | 0.4964 | 0.6775 | 0.8912 | 0.0009 *** | 0.0268 * | 0.4423 |
| China | 0.0948 | 0.0870 | 0.4707 | 0.7309 | 0.7335 | 0.0995 |
| Russia | 0.6297 | 0.5957 | 0.0577 | 0.8246 | 0.6897 | 0.2555 |
| Brazil | 0.2648 | 0.0435 * | 0.3470 | 0.1724 | 0.1267 | 0.3427 |
| India | 0.0074 ** | 0.2449 | 0.3934 | 0.8505 | 0.5818 | 0.2473 |
| South Africa | 0.0047 ** | 0.0022 ** | 0.1917 | 0.0094 ** | 0.0615 | 0.9620 |

Note: For Debt → Loan, H0: Debt does not cause Loan. ***, **, and * denote rejection of the null hypothesis at the 1, 5, and 10% significance level, respectively.

    Although it is difficult to find commonality of the impact of sovereign debt on bank NPLs across countries, we note that countries with high sovereign debt ratio are usually associated with higher NPLs. An increase in NPLs reflects the deterioration of banks' balance sheets and asset quality, which in turn may reduce banks' leverage or profits. Losses on government bonds weaken banks' balance sheets and increase financing costs. Meanwhile, countries with larger amounts of sovereign debt exhibit a one-way causal relationship between bank NPLs and sovereign debt or bi-directional causality. It is noteworthy that the different characteristics across countries and the heterogeneity in the results point towards caution when making inferences about the relationship between sovereign debt and NPLs.

### 4.2. Kendall's Tau Results

Kendall's tau shows the correlation between each pair of sample countries. We categorize all pairs of countries into two groups according to the geographical location: European countries and other (BRICS) countries. Table 5 reports the means of Kendall's tau between the government-to-GDP ratio and the bank NPLs for the sample countries. First, within the European group, we calculate the means of Kendall's tau between each country and the other countries and report the results in Panel A. Since sovereign debt crisis occurs in European countries, to easily understand the spillover effect of sovereign debt crisis on the emerging countries, we calculate the means of Kendall's tau between each BRICS country and all European countries and report the results in Panel B. We find that the means of tau within European countries are much higher than the means within BRICS countries.

**Table 5.** Mean of Kendall's tau for the NPLs ratio and the government-to-GDP ratio.

| | Mean of NPLs | Mean of Debt |
|---|---|---|
| Panel A: Kendall's Tau within European Countries | | |
| 25 EU countries | 0.4742 | 0.5815 |
| Belgium-24 EU countries | 0.4390 | 0.6167 |
| Bulgaria-24 EU countries | 0.4695 | 0.3874 |
| Czech Republic-24 EU countries | 0.4292 | 0.4951 |
| Denmark-24 EU countries | 0.4475 | 0.3667 |
| Germany-24 EU countries | −0.0679 | 0.2402 |
| Estonia-24 EU countries | 0.2619 | 0.5377 |
| Ireland-24 EU countries | 0.4351 | 0.4182 |
| Greece-24 EU countries | 0.2847 | 0.5571 |
| Spain-24 EU countries | 0.4592 | 0.6435 |
| France-24 EU countries | 0.4425 | 0.5350 |
| Croatia-24 EU countries | 0.4328 | 0.6147 |
| Italy-24 EU countries | 0.3719 | 0.5701 |
| Cyprus-24 EU countries | 0.3231 | 0.5619 |
| Latvia-24 EU countries | 0.3388 | 0.3293 |
| Lithuania-24 EU countries | 0.2956 | 0.5814 |
| Hungary-24 EU countries | 0.4758 | 0.2210 |
| Malta-24 EU countries | 0.3593 | 0.0442 |
| Austria-24 EU countries | 0.3469 | 0.4761 |
| Poland-24 EU countries | 0.0438 | 0.4184 |
| Portugal-24 EU countries | 0.2968 | 0.6035 |
| Romania-24 EU countries | 0.4594 | 0.6086 |
| Slovenia-24 EU countries | 0.4517 | 0.5915 |
| Slovakia-24 EU countries | 0.3841 | 0.5610 |
| Sweden-24 EU countries | 0.1051 | 0.2437 |
| United Kingdom-24 EU countries | 0.2969 | 0.5610 |
| Panel B: Kendall's Tau between BRICS and European Countries | | |
| 5 BRICS countries | 0.0141 | 0.2582 |
| China-25 EU countries | −0.3992 | 0.4438 |
| Russia-25 EU countries | 0.1456 | 0.5874 |
| Brazil-25 EU countries | −0.0064 | 0.0387 |
| India-25 EU countries | 0.0700 | −0.3436 |
| South Africa-25 EU countries | 0.2498 | 0.5006 |

This result is not surprising because there are strong commonalities among European countries, such as currency, geographical location, culture, etc. In particular, the highest means of tau occur in Hungary and Bulgaria, in contrast to other European countries. While in BRICS countries, although the economy grows relatively fast in these countries, the business models and the engines of economy are very different, which leads to a low mean of correlations among these countries. Moreover, we note that the means of tau between each BRICS country and the European countries varies significantly.

For example, the mean of correlation between South Africa and the European countries is positive but the mean between Brazil and the European countries is negative. The heterogeneity of correlations across countries might be due to the variation of country characteristics, which is out of the scope of this paper. Sovereign risk has a negative spillover effect on bank risk, and failure to fully protect the banking system from the impact of serious domestic sovereignty is a reason to maintain good public finances.

### 4.3. Copula Results

We consider three copula models to analyze the tail dependence between the government-to-GDP ratio and the bank non-performing loan ratio: Student's *t* copula (symmetric association of tail dependence), rotated Clayton copula (upper-tail dependence), and Joe copula (upper-tail dependence). In terms of tail dependence, these series copula models cover the major combinations of features necessary to capture possible associations between the variables studied, and they are the most commonly used copulas in finance [7].

The estimation results of the three copula models above are shown in Table 6. The *t*-copula detects both upper and lower tail dependence at each sample country. Because of the symmetry of the Student's *t* distribution, the upper tail coefficient is generally equal to the lower tail coefficient. We find that the highest dependence of upper tails occurs in Ireland, about 0.8778. The large and positive tail dependence suggests a strong correlation between the extreme expansion of sovereign debt and the sharp increase of bank NPLs. In addition, there are five countries in which the upper tail dependence is greater than 0.5, including Denmark, Ireland, Croatia, Latvia, and Portugal.

**Table 6.** Tail dependence for different copulas with the NPLs ratio and the government-to-GDP ratio.

| Country | t Copula | | Rotated Clayton Copula | | Joe Copula | |
|---|---|---|---|---|---|---|
| | Lower | Upper | Lower | Upper | Lower | Upper |
| Belgium | 0.4045 | 0.4045 | 0 | 0.8047 | 0 | 0.8115 |
| Bulgaria | 0.0015 | 0.0015 | 0 | 0.3241 | 0 | 0.3892 |
| Czech Republic | 0.0917 | 0.0917 | 0 | 0.5971 | 0 | 0.6215 |
| Denmark | 0.7460 | 0.7460 | 0 | 0.7905 | 0 | 0.8002 |
| Germany | 0.0011 | 0.0011 | 0 | 0.2882 | 0 | 0.3599 |
| Estonia | 0.0001 | 0.0001 | 0 | 0.0000 | 0 | 0.0001 |
| Ireland | 0.8778 | 0.8778 | 0 | 0.9231 | 0 | 0.9237 |
| Greece | 0.2977 | 0.2977 | 0 | 0.7800 | 0 | 0.7887 |
| Spain | 0.1112 | 0.1112 | 0 | 0.6620 | 0 | 0.6751 |
| France | 0.1909 | 0.1909 | 0 | 0.1300 | 0 | 0.1185 |
| Croatia | 0.8252 | 0.8252 | 0 | 0.9011 | 0 | 0.9020 |
| Italy | 0.2979 | 0.2979 | 0 | 0.7379 | 0 | 0.7514 |
| Cyprus | 0.2074 | 0.2074 | 0 | 0.8356 | 0 | 0.8412 |
| Latvia | 0.6543 | 0.6543 | 0 | 0.7781 | 0 | 0.7830 |
| Hungary | 0.0620 | 0.0620 | 0 | 0.5273 | 0 | 0.5548 |
| Malta | 0.4359 | 0.4359 | 0 | 0.5465 | 0 | 0.5850 |
| Austria | 0.0817 | 0.0817 | 0 | 0.7646 | 0 | 0.7698 |
| Poland | 0.1781 | 0.1781 | 0 | 0.0000 | 0 | 0.0001 |
| Portugal | 0.7246 | 0.7246 | 0 | 0.7089 | 0 | 0.7266 |
| Romania | 0.1218 | 0.1218 | 0 | 0.6658 | 0 | 0.6762 |
| Slovenia | 0.0083 | 0.0083 | 0 | 0.3696 | 0 | 0.4003 |
| Slovakia | 0.0057 | 0.0057 | 0 | 0.2229 | 0 | 0.2468 |
| Sweden | 0.0096 | 0.0096 | 0 | 0.5883 | 0 | 0.6172 |
| Russia | 0.0048 | 0.0048 | 0 | 0.2807 | 0 | 0.3127 |
| Brazil | 0.0043 | 0.0043 | 0 | 0.5227 | 0 | 0.5571 |

Then, we use the rotated Clayton and Joe copulas as alternative methods to examine the upper tail dependence between sovereign debt ratio and bank NPLs ratio. In contrast to the Student's *t*

distribution, the underlying distributions in the rotated Clayton copula and Joe copula focus on the upper tail dependence. According to Table 6, we note that the tail dependence levels in all countries are consistent in the rotated Clayton and Joe copulas but very different from the dependence in the *t* copula. The highest upper tail dependence occurs in Ireland as well under both the rotated Clayton and Joe copulas. We document 17 out of 25 countries whose value of upper tail dependence is greater than 0.5 using either the rotated Clayton copula or Joe copula which are symmetric copula functions so that those Archimedean copula functions cannot fit asymmetric distributed data well.

*4.4. Gaussian Copula Regression Method Results*

In previous sections, we focused on examining the causality using statistical methods without controlling for the other determinants of bank NPLs. Related literature identifies many factors that drive bank NPLs ratios, at both macro- and micro-levels [14–17,48]. To isolate the impact of other known determinants, we employ the Gaussian copula regression method (GCRM) in this section. Since we use the aggregated level of bank NPLs over total loans, we focus on the macroeconomic variables. Specifically, we control for GDP, inflation rate, government fiscal expenditure, and government fiscal revenue in Gaussian copula regressions.

Because of heterogeneity across countries, as shown in the previous analysis, we perform the Gaussian copula regressions country by country and report regression results in Table 7. We find positive and significant coefficients for sovereign debt ratio in 17 out of 25 countries, including Cyprus, Greece, Croatia, Malta, Italy, Romania, Portugal, Slovenia, Spain, Ireland, Belgium, Germany, Hungary, Czech Republic, Sweden, Slovakia, and Brazil. These positive and significant coefficients in a majority of the countries provide further support for the positive impact of sovereign debt ratio on bank NPLs ratios. This positive relationship highlights that fiscal stress could be a potential factor that deteriorates bank loan performance. In addition, we also document negative and significant coefficients in several countries including France, the United Kingdom, and South Africa.

Regarding the control variables, their coefficients vary across countries. First, the impact of the GDP growth rate on bank NPLs ratios is mixed. On one hand, positive economic growth for each economy indicates an increase in the wealth of private sector individuals, enterprises, and other institutions, which results in a strong capability of repaying their respective debts and a decrease of bank NPLs ratios. On the other hand, the expansion of the economy is usually associated with credit booming. For instance, the credit bubble before the sub-prime financial crisis. The cheap credit during the expansion of the economy sows the seeds for non-performing loans, which suggests a positive relation between GDP growth and bank NPLs ratio. According to the results in Table 7, we find positive and negative coefficients for GDP growth rate in 7 and 4 countries, respectively, while the coefficients of GDP growth rate are insignificant in other countries.

Second, a high inflation rate is usually accompanied by an expansionary monetary policy. The enlarged monetary base under an expansionary monetary policy increases the supply of loans. Under such circumstance, banks are more likely to adopt an aggressive strategy for lending, which possibly results in a higher level of non-performing loans. Thus, we expect a positive relationship between inflation rate and bank NPL ratio. However, we only document significant and positive coefficients for inflation rate in five countries. On the contrary, we find significantly negative coefficients for inflation in eight countries. One of the possible explanations is that an increase in economic activity leads to enhanced demand for loans, which in turn can cause higher lending rates. In addition, increased economic activity can reduce defaults and increase deposits because it can make business more profitable. However, tightening monetary policy can increase interest rates and make banks be more inclined to attract customers with higher risks and compensate them for the high risk by raising loan interest rates. These results are consistent with the studies by Were and Wambua [49] and Ghosh [27].

**Table 7.** GCMR estimation results of the NPLs ratio to the government-to-GDP ratio.

| Country | Intercept | Debt | Expenditure | GDP | Revenue | Inflation |
|---|---|---|---|---|---|---|
| Bulgaria | 44.8120 * | −0.2404 | 0.4717 | 0.5576 | −1.2776 * | −1.2742 * |
| | (18.9509) | (0.3626) | (0.5271) | (0.4915) | (0.6131) | (0.6076) |
| Cyprus | 1.7552 *** | 0.7646 *** | −0.7424 | 0.1576 | −0.2369 | −0.5485 |
| | (0.0024) | (0.0715) | (0.5010) | (0.5022) | (0.5877) | (0.8323) |
| Croatia | 53.0771 *** | 0.2805 *** | 0.3155 *** | 0.0338 | −1.0389 *** | −0.2666 *** |
| | (0.0004) | (0.0093) | (0.0562) | (0.0412) | (0.062) | (0.07863) |
| Greece | −118.6 *** | 0.2387 ** | 0.5013 | 1.2320 ** | 1.8230 *** | −0.3670 |
| | (0.0065) | (0.0903) | (0.2686) | (0.3769) | (0.5232) | (0.6392) |
| Italy | −76.6442 *** | 0.22266 *** | 0.2701 | 0.2415 * | 1.0594 *** | −0.3310 |
| | (0.0014) | (0.02853) | (0.2545) | (0.1127) | (0.2863) | (0.2282) |
| Malta | 28.7748 * | 0.3301 *** | −0.0692 | 0.3674 *** | 0.4126 | −0.1790 |
| | (11.5029) | (0.0796) | (0.2348) | (0.0684) | (0.3108) | (0.2387) |
| Portugal | −4.1232 | 0.1813 *** | 0.3222 * | 0.0537 | 0.1857 | −0.0917 |
| | (12.4124) | (0.0146) | (0.1385) | (0.1089) | (0.2245) | (0.1744) |
| Romania | −12.5478 | 0.6090 *** | −1.1359 | −0.3779 | 1.3277 * | 1.1883 ** |
| | (20.7407) | (0.0893) | (0.8139) | (0.2783) | (0.5304) | (0.4413) |
| Spain | −27.9760 ** | 0.0699 *** | 0.3601 | −0.2669 | 0.3274 * | 0.0860 |
| | (10.5332) | (0.0135) | (0.1946) | (0.1958) | (0.1546) | (0.1712) |
| Slovenia | −131.3625 *** | 0.1401 *** | 0.1843 | −0.4206 ** | 3.0411 *** | 0.8356 |
| | (21.3408) | (0.033) | (0.1469) | (0.1370) | (0.5862) | (0.4637) |
| Belgium | −19.6475 *** | 0.04560 *** | 0.2127 *** | 0.0205 | 0.1381 ** | −0.1122 *** |
| | (0.9114) | (0.0096) | (0.0235) | (0.0221) | (0.0279) | (0.0212) |
| France | −42.1100 *** | −0.0807 *** | 0.7195 *** | 0.0410 | 0.2384 *** | 0.0857 |
| | (0.0005) | (0.0096) | (0.0678) | (0.0713) | (0.0697) | (0.0971) |
| Ireland | −9.76100 | 0.2519 *** | −0.2364 *** | 0.0107 | 0.4172 * | −1.0583 * |
| | (5.3245) | (0.0139) | (0.0689) | (0.0699) | (0.2016) | (0.4138) |
| United Kingdom | −35.2600 *** | −0.0104 * | 0.6023 *** | 0.0595 | 0.3763 *** | 0.00591 |
| | (0.0009) | (0.0051) | (0.0418) | (0.0528) | (0.0433) | (0.0802) |
| Austria | −38.1600 *** | 0.0000 | 0.2473 *** | 0.2690 *** | 0.5744 *** | −0.0910 *** |
| | (0.0003) | (0.9981) | (0.0000) | (0.0275) | (0.0412) | (0.0469) |
| Czech Republic | 23.0463 *** | 0.2120 *** | 0.0625 | 0.0931 *** | −0.7161 *** | −0.2335 *** |
| | (0.0009) | (0.0147) | (0.0389) | (0.0281) | (0.0456) | (0.0445) |
| Germany | 18.3133 *** | 0.0187 * | 0.1316 *** | −0.0854 *** | −0.5232 *** | 0.2074 ** |
| | (0.0004) | (0.0090) | (0.0363) | (0.0225 | (0.0312) | (0.0780) |
| Hungary | −155.4512 *** | 0.8237 *** | 1.1118 * | 0.5173 | 1.0447 | 0.1635 |
| | (40.5536) | (0.2018) | (0.5583) | (0.3618) | (0.6036) | (0.4481) |
| Poland | −49.4549 * | 0.1730 | 0.5021* | 0.0908 | 0.6117 | −3.7770 * |
| | (19.4889) | (0.1054) | (0.1993) | (0.2431) | (0.3446) | (0.1628) |
| Slovakia | −4.06700 | 0.0981 *** | 0.4829 *** | 0.0273 | −0.4233 *** | 0.1052 |
| | (3.3320) | (0.0171) | (0.0775) | (0.0385) | (0.0788) | (0.0874) |
| Denmark | −21.8500 * | 0.0696 | 0.4173 * | 0.0754 | −0.0008 | −0.1979 |
| | (1094) | (0.0787) | (0.1649) | (0.0830) | (0.1549) | (0.1464) |
| Sweden | −0.0660 | 0.1284 *** | 0.0373 | −0.0792 *** | −0.1271 * | 0.3247 *** |
| | (3.0839) | (0.0226) | (0.0594) | (0.0212) | (0.0508) | (0.0950) |
| Estonia | −48.8265 ** | 0.0166 | −0.4439 | −0.0130 | 1.6916 | 0.5242 |
| | (18.7886) | (0.3587) | (0.7987) | (0.1615) | (1.0694) | (0.3737) |
| Latvia | −212.5658 *** | −0.1794 | 3.2445 *** | 0.6411 *** | 2.7942 ** | 0.6060 *** |
| | (39.9114) | (0.0922) | (0.3430) | (0.1166) | (0.9595) | (0.1792) |
| Lithuania | 12.7869 | 0.1462 | 2.0778 *** | −0.1690 | −2.4429 * | −0.3245 |
| | (35.3126) | (0.1060) | (0.3039) | (0.1391) | (0.9685) | (0.3397) |
| China | 14.7477 | 0.1116 | 0.1928 | −0.0441 | −0.8561 * | 0.08932 |
| | (8.7543) | (0.1435) | (0.5110) | (0.4064) | (0.3772) | (0.1556) |
| Russia | 43.9618 | −0.1486 | 0.0552 | −0.1515 | −1.0344 ** | −0.1898 |
| | (25.632) | (0.1724) | (0.3749) | (0.1686) | (0.3970) | (0.1067) |
| Brazil | −7.6525 | 0.0515 * | 0.0048 | −0.1513 ** | 0.2473 ** | −0.0237 |
| | (4.7255) | (0.0227) | (0.1067) | (0.0466) | (0.0774) | (0.0685) |
| India | 1.2336 | −0.2745 | 1.1337 | −0.0997 | −0.0520 | −0.8655 *** |
| | (20.9252) | (0.1432) | (0.61111) | (0.2197) | (−0.5827) | (0.1773) |
| South Africa | 4.9088 *** | −0.133 *** | 1.0070 *** | 0.2845 ** | −1.0491 *** | 0.2123 * |
| | (0.0009) | (0.0188) | (0.0887) | (0.1061) | (0.1153) | (0.0845) |

Note: * Significance at 10% level. ** Significance at 5% level. *** Significance at 1% level. Standard errors are in parentheses.

Government fiscal expenditure and revenue is found to be associated with NPLs. More specifically, the ratio of government fiscal expenditure is negatively correlated with NPLs in Croatia, Portugal, and Ireland and indicates that the rise in government fiscal expenditure decreases the NPLs growth. In contrast, the coefficients for expenditure are positive and statistically significant in Belgium, France, United Kingdom, Austria, Germany, Hungary, Poland, Slovakia, Denmark, Latvia, Lithuania, and South Africa. The implication of these results is that an increase in expenditure stimulates the NPLs growth in the long run. Although the flow of government fiscal expenditure improves productivity, the government should not borrow money to fund it because the resulting increase in public debt would reduce welfare and growth rates. Similarly, there is a significant negative relationship between government fiscal revenue and NPLs in Bulgaria, Croatia, Czech Republic, Germany, Slovakia, Sweden, Lithuania, China, Russia, and South Africa. It can be argued that governments are challenged in the capital markets when accessing loanable funds where the demand for loanable funds far exceeds its supply. Consequently, government fiscal revenue increases capital cost and default risk which increases NPLs. In contrast, revenue is positively significant in Greece, Italy, Romania, Spain, Slovenia, and so on, most of which belong to South and Western Europe. Government fiscal revenue increases (falls) and government spending falls (increases) in good (bad) times. An increase in GDP brings about an increase in government fiscal revenue, which in turn raises the payment capacity of the government and hence reduces NPLs. This relationship highlights that the fiscal problems might lead to a substantial increase in problem loans.

## 5. Conclusions

In this study, we investigate the tail dependence between sovereign debt distress and NPLs using a large sample of developed and emerging countries in recent decades. This paper covers the period of 11 years (2006–2017), which includes both the global financial crisis and European sovereign debt crisis period. Some meaningful results are obtained. The results may have some meaningful implications for policymakers because unsustainable sovereign debt can lead to payment defaults, which will impose more problems on the stability of the region.

According to Granger causality tests, we found a heterogeneity of causality between sovereign debt and bank NPLs across countries. In the majority of countries, we documented a significant causality between government-to-GDP ratio and bank NPLs ratio. We note that countries with a high sovereign debt ratio are usually associated with higher NPLs. An increase in NPLs reflects the deterioration of banks' balance sheets and asset quality, which in turn may reduce banks' leverage or profits. Meanwhile, countries with larger amounts of sovereign debt exhibit a one-way causal relationship between bank NPLs and sovereign debt or bi-directional causality.

Then, we used Kendall's tau as an alternative measure to examine the correlation of sovereign debt and bank NPLs in each pair of countries. We found that the internal correlations within European countries are much higher than within the BRICS countries. This result is not surprising because there are strong commonalities among European countries, the highest means of tau occur in Hungary and Bulgaria. In BRICs countries, although their economic growth is relatively fast, the business models and the engines of economy are very different, resulting in a low average level of correlation between these countries. Moreover, we note that the means of tau between each BRICS country and the European countries varies significantly.

Next, we employed three copula functions to investigate the upper tail dependence between bank NPLs and sovereign debt distress. We found a significant variation of tail dependence across countries. The large and positive tail dependence suggests a strong correlation between the extreme expansion of sovereign debt and the sharp increase of bank NPLs. The higher tail dependence coefficients of these countries imply that bank NPLs is more vulnerable to the expansion of sovereign debt during our sample period.

Finally, we used a Gaussian copula regression method to isolate the impact of sovereign debt ratio on bank NPL ratio by controlling for government expenditure, revenue, GDP growth rate and

inflation rate. We found a significant and positive relation between sovereign debt ratio and bank NPLs in most of the countries, which adds credence to the positive impact of sovereign debt distress on bank loan crises. We have evidence of the effects of macroeconomic determinants on the increase of NPLs. These results are consistent with Reinhart and Rogoff's [8] empirical studies that bank crises are usually accompanied by a large increase in sovereign debt. Furthermore, government spending tends to be greater than revenues when a financial crisis occurs [50].

This study contributes to the financial literature by investigating the tail dependence between sovereign debt distress and bank NPLs. The results of this study confirm that the contribution of countries with large sovereign debt scale to NPLs has increased significantly since 2008, especially for EU countries. This means that the expansion of sovereign debt by banks is the main determinant of bank NPLs. These results can help financial institutions find out which countries' debts may not be sustainable and which countries' final payment default may bring more problems to the stability of the region. These analyses could also help regulators who are trying to understand the relationship between sovereign debt and bank NPLs, as well as financial institutions that may hold large amounts of sovereign debt.

Therefore, macroprudential policies and sound regulation are also crucial, as a strong strict credit and capital base and liquidity risk management practices are essential to curb the impact of sovereign tensions on banks. Furthermore, the results of this study can also help to establish a better regulatory mechanism and ultimately punish the countries that violate the rules. This study can be expanded by including bank-specific data and macroeconomic variables over a longer period of time. In addition, the examined model could be applied to other developed and underdeveloped countries in addition to the EU and BRICS countries, putting forward a more comprehensive view of factors that affect the NPLs.

**Author Contributions:** L.L. and Y.-M.L. designed this research and the model, analyzed the data and wrote the paper. J.-M.K., R.Z., and G.-Q.R. obtained inference, analyzed the data, and provided editorial supports. All the authors cooperated to revise the paper. All authors have read and agreed to the published version of the manuscript.

**Funding:** This research was funded by National Natural Science Foundation of China Grant (nos. U1904211, 71702171, 71672182, U1604262).

**Conflicts of Interest:** The authors declare no conflict of interest.

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
