# Peer review of "Analysis of Tail Dependence between Sovereign Debt Distress and Bank Non-Performing Loans"

_sustainability, doi:10.3390/su12020747_

Round 1

Reviewer 1 Report

This is a good, well written and articulated paper.

What might be improved is the title (which should be more specific like, for instance, by adding a subtitle) and the abstract. Otherwise stated, the authors should better highlight that the following is a concrete, empirically based analysis with direct policy implications.

Author Response

First of all, we’d like to express our gratitude to the referees for their careful and critical reading of our manuscript. 

As described in a reply below to the reviewers, we've revised our manuscript following suggestions line by line.

Overall, our revised manuscript has been considerably improved due to the revision we've made according to the reviewers' suggestions. 

We hope that our revised manuscript would be suitable for the publication in the Sustainability.

Point: This is a good, well written and articulated paper.

What might be improved is the title (which should be more specific like, for instance, by adding a subtitle) and the abstract. Otherwise stated, the authors should better highlight that the following is a concrete, empirically based analysis with direct policy implications.

Response 1: According to the referee’s valuable comment, new title will be revised as "Analysis of Tail Dependence between Sovereign Debt Distress and Bank Non-performing Loans"

Response 2: According to the referee’s valuable comment, the conclusion is revised adding the paragraph below:

In this study, we investigate the tail dependence between sovereign debt distress and NPLs using a large sample of developed and emerging countries in recent decades. This paper covers the period of 11 years (2006-2017), which includes both the global financial crisis and European sovereign debt crisis period. Some meaningful results are obtained. The results may have some meaningful implications for policymakers because unsustainable sovereign debt can lead to payment defaults, which will impose more problems on the stability of the region.

According to Granger causality tests, we found a heterogeneity of causality between sovereign debt and bank NPLs across countries. In the majority of countries, we documented a significant causality between government-to-GDP ratio and bank NPLs ratio. We note that countries with high sovereign debt ratio are usually associated with higher NPLs. An increase in NPLs reflects the deterioration of banks’ balance sheets and asset quality, which in turn may reduce banks' leverage or profits.  Meanwhile, countries with larger amounts of sovereign debt exhibit a one-way causal relationship between bank NPLs and sovereign debt or bi-directional causality.

Then, we used Kendall’s tau as an alternative measure to examine the correlation of sovereign debt and bank NPLs in each pair of countries. We found that the internal correlations within European countries are much higher than within the BRICS countries. This result is not surprising because there are strong commonalities among European countries, the highest means of tau occur in Hungary and Bulgaria. In BRICs countries, although their economic growth is relatively fast, the business models and the engines of economy are very different, resulting in a low average level of correlation between these countries. Moreover, we note that the means of tau between each BRICS country and the European countries varies significantly.

Next, we employed three copula functions to investigate the upper tail dependence between bank NPLs and sovereign debt distress. We found a significant variation of tail dependence across countries. The large and positive tail dependence suggests a strong correlation between the extreme expansion of sovereign debt and the sharp increase of bank NPLs. The higher tail dependence coefficients of these countries imply that bank NPLs is more vulnerable to the expansion of sovereign debt during our sample period.

Finally, we used a Gaussian copula regression method to isolate the impact of sovereign debt ratio on bank NPL ratio by controlling for government expenditure, revenue, GDP growth rate and inflation rate. We found a significant and positive relation between sovereign debt ratio and bank NPLs in most of the countries, which adds credence to the positive impact of sovereign debt distress on bank loan crises. We have evidence of the effects of macroeconomic determinants on the increase of NPLs. These results are consistent with Reinhart and Rogoff's [8] empirical studies that bank crises are usually accompanied by a large increase in sovereign debt. Furthermore, government spending tends to be greater than revenues when a financial crisis occurs [50].

This study contributes to the financial literature by investigating the tail dependence between sovereign debt distress and bank NPLs. The results of this study confirm that the contribution of countries with large sovereign debt scale to NPLs has increased significantly since 2008, especially for EU countries. This means that the expansion of sovereign debt by banks is the main determinant of bank NPLs. These results can help financial institutions find out which countries' debts may not be sustainable and which countries' final payment default may bring more problems to the stability of the region. These analyses ccould also help regulators who are trying to understand the relationship between sovereign debt and bank NPLs, as well as financial institutions that may hold large amounts of sovereign debt.

Therfore, macroprudential policies and sound regulation are also crucial, as a strong strict credit and capital base and liquidity risk management practices are essential to curb the impact of sovereign tensions on banks. Furthermore, the results of this study can also help to establish a better regulatory mechanism and ultimately punish the countries that violate the rules. This study can be expanded by including bank-specific data and macroeconomic variables over a longer period of time. In addition, the examined model could be applied to other developed and underdeveloped countries in addition to the EU and BRICS countries, putting forward a more comprehensive view of factors that affect the NPLs.

Finally, we really appreciate the referees’ critical but valuable comments which have been guidance in revising our manuscript. We believe that our manuscript has been much improved due to the revision based on the referees’ suggestions.

Reviewer 2 Report

The strengths of the article is original and interesting considerations with is consistent with the pattern of research. Solid methodology of the research with statistical analysis. Therefore contribution to existing knowledge is considerable.

The weakness of this article is too little the conclusions and are some problems with organization & readability. For example:

authors can improve the quality of figures (I understand that there are a lot of lines of analyzed countries, therefore for clarity I suggest introducing additional markings in the form of dots, circles, crosses, etc.) In addition, similar colors of several countries may give the impression of ranking countries according to a certain feature. I also propose to extend the conclusion regarding the obtained research results and refer to the results obtained by other researchers cited in this article. improve the citation of scientific papers (formatting method and some mistakes, for example for items: "Alter, A., A. Beyer., 2014. The dynamics of spillover effects during the European sovereign debt turmoil. Journal of Banking & Finance 42, 134-153" - set the formatting as for other literature sources and correct the date, because the text is 2013 year, but in the list of references is 2014 year).

Without few suggestions for improving the content I congratulations for author in taking an effort in writing this manuscript, because in generally it is excellent article and very good original and interesting considerations, which is consistent with the pattern of research. It is model article worthy of imitation.

Author Response

First of all, we’d like to express our gratitude to the referees for their careful and critical reading of our manuscript. 

As described in a reply below to the reviewers, we've revised our manuscript following suggestions line by line.

Overall, our revised manuscript has been considerably improved due to the revision we've made according to the reviewers' suggestions. 

We hope that our revised manuscript would be suitable for the publication in the Sustainability.

Point: The strengths of the article is original and interesting considerations with is consistent with the pattern of research. Solid methodology of the research with statistical analysis. Therefore contribution to existing knowledge is considerable.

The weakness of this article is too little the conclusions and are some problems with organization & readability. For example:

authors can improve the quality of figures (I understand that there are a lot of lines of analyzed countries, therefore for clarity I suggest introducing additional markings in the form of dots, circles, crosses, etc.) In addition, similar colors of several countries may give the impression of ranking countries according to a certain feature. I also propose to extend the conclusion regarding the obtained research results and refer to the results obtained by other researchers cited in this article. improve the citation of scientific papers (formatting method and some mistakes, for example for items: "Alter, A., A. Beyer., 2014. The dynamics of spillover effects during the European sovereign debt turmoil. Journal of Banking & Finance 42, 134-153" - set the formatting as for other literature sources and correct the date, because the text is 2013 year, but in the list of references is 2014 year).

Without few suggestions for improving the content I congratulations for author in taking an effort in writing this manuscript, because in generally it is excellent article and very good original and interesting considerations, which is consistent with the pattern of research. It is model article worthy of imitation.

Response 1: According to the referee’s valuable comment, we introduce additional markings in the form of dots, circles, crosses, etc. for clarity, see the new figures 1 and 2 in the revised version.

Response 2: According to the referee’s valuable comment, the conclusion is revised adding the paragraph below:

In this study, we investigate the tail dependence between sovereign debt distress and NPLs using a large sample of developed and emerging countries in recent decades. This paper covers the period of 11 years (2006-2017), which includes both the global financial crisis and European sovereign debt crisis period. Some meaningful results are obtained. The results may have some meaningful implications for policymakers because unsustainable sovereign debt can lead to payment defaults, which will impose more problems on the stability of the region.

According to Granger causality tests, we found a heterogeneity of causality between sovereign debt and bank NPLs across countries. In the majority of countries, we documented a significant causality between government-to-GDP ratio and bank NPLs ratio. We note that countries with high sovereign debt ratio are usually associated with higher NPLs. An increase in NPLs reflects the deterioration of banks’ balance sheets and asset quality, which in turn may reduce banks' leverage or profits.  Meanwhile, countries with larger amounts of sovereign debt exhibit a one-way causal relationship between bank NPLs and sovereign debt or bi-directional causality.

Then, we used Kendall’s tau as an alternative measure to examine the correlation of sovereign debt and bank NPLs in each pair of countries. We found that the internal correlations within European countries are much higher than within the BRICS countries. This result is not surprising because there are strong commonalities among European countries, the highest means of tau occur in Hungary and Bulgaria. In BRICs countries, although their economic growth is relatively fast, the business models and the engines of economy are very different, resulting in a low average level of correlation between these countries. Moreover, we note that the means of tau between each BRICS country and the European countries varies significantly.

Next, we employed three copula functions to investigate the upper tail dependence between bank NPLs and sovereign debt distress. We found a significant variation of tail dependence across countries. The large and positive tail dependence suggests a strong correlation between the extreme expansion of sovereign debt and the sharp increase of bank NPLs. The higher tail dependence coefficients of these countries imply that bank NPLs is more vulnerable to the expansion of sovereign debt during our sample period.

Finally, we used a Gaussian copula regression method to isolate the impact of sovereign debt ratio on bank NPL ratio by controlling for government expenditure, revenue, GDP growth rate and inflation rate. We found a significant and positive relation between sovereign debt ratio and bank NPLs in most of the countries, which adds credence to the positive impact of sovereign debt distress on bank loan crises. We have evidence of the effects of macroeconomic determinants on the increase of NPLs. These results are consistent with Reinhart and Rogoff's [8] empirical studies that bank crises are usually accompanied by a large increase in sovereign debt. Furthermore, government spending tends to be greater than revenues when a financial crisis occurs [50].

This study contributes to the financial literature by investigating the tail dependence between sovereign debt distress and bank NPLs. The results of this study confirm that the contribution of countries with large sovereign debt scale to NPLs has increased significantly since 2008, especially for EU countries. This means that the expansion of sovereign debt by banks is the main determinant of bank NPLs. These results can help financial institutions find out which countries' debts may not be sustainable and which countries' final payment default may bring more problems to the stability of the region. These analyses ccould also help regulators who are trying to understand the relationship between sovereign debt and bank NPLs, as well as financial institutions that may hold large amounts of sovereign debt.

Therfore, macroprudential policies and sound regulation are also crucial, as a strong strict credit and capital base and liquidity risk management practices are essential to curb the impact of sovereign tensions on banks. Furthermore, the results of this study can also help to establish a better regulatory mechanism and ultimately punish the countries that violate the rules. This study can be expanded by including bank-specific data and macroeconomic variables over a longer period of time. In addition, the examined model could be applied to other developed and underdeveloped countries in addition to the EU and BRICS countries, putting forward a more comprehensive view of factors that affect the NPLs.

Response 3: This is a mistake in writing, we correct the date according literature. we examine other literature format again to make sure they are correct.

Finally, we really appreciate the referees’ critical but valuable comments which have been guidance in revising our manuscript. We believe that our manuscript has been much improved due to the revision based on the referees’ suggestions.
